# Validity and Reliability of a Questionnaire on Attitudes, Knowledge, and Perceptions of Pharmacy Students Regarding the Training Received on Antibiotics and Antimicrobial Resistance during Their University Studies

**DOI:** 10.3390/antibiotics13090811

**Published:** 2024-08-26

**Authors:** Patricia Otero-Batán, Guillermo Lens-Perol, Olalla Vázquez-Cancela, Angel Salgado-Barreira, Juan Manuel Vazquez-Lago

**Affiliations:** 1Department of Preventive Medicine and Public Health Service, University Hospital of Santiago de Compostela, Rua da Choupana s/n, 15705 Santiago de Compostela, Spain; 2Health Research Institute of Santiago de Compostela (IDIS), 15706 Santiago de Compostela, Spain; 3Department of Preventive Medicine and Public Health Service, Faculty of Pharmacy, University of Santiago de Compostela, Campus Vida s/n, 15705 Santiago de Compostela, Spain

**Keywords:** validity, reliability, attitudes, knowledge, antimicrobial resistances

## Abstract

Background: Antimicrobial resistance is a major public health issue today. Therefore, it is essential to focus on the education of pharmacists as future dispensers. The objective of this study was to validate a questionnaire that assesses the knowledge, attitudes, and perceptions of pharmacy students regarding the education received during their university degree on the use and dispensation of antibiotics, as well as bacterial resistance. Methods: An online questionnaire was developed and distributed via RedCap v.13.7.1 to pharmacy students at the University of Santiago de Compostela using the WhatsApp social network. The questionnaire consisted of 28 items evaluating 5 dimensions: “quality of care”, “communication skills”, “antibiotic resistance”, “teaching methodology”, and “education on antibiotics at the faculty”. The questionnaire validation was conducted in 2 steps: Step 1 involved content and appearance validation, and Step 2 involved reliability analysis. Results: A total of 61 completed questionnaires were received. The mean age was 21.82 ± 3.81 years, with 20 males (32.8%) and 41 females (67.2%). Content validation was performed through a nominal group of 5 experts, and appearance validation was conducted by a focus group of 6 university pharmacy students. The questionnaire demonstrated a Cronbach’s alpha value of 0.80 and adequate item discrimination capability. Confirmatory factor analysis was performed to assess construct validity, confirming the 5 predefined dimensions. Conclusions: A questionnaire has been developed and validated with high reliability and validity. Its use will help identify areas for improvement in the university education of pharmacy students, ultimately contributing to better use and dispensation of antibiotics and thereby improving antimicrobial resistance.

## 1. Introduction

Antibiotics are drugs used in the treatment of bacterial infections that have saved countless lives since their discovery [1]. However, the increasing bacterial resistance to these drugs poses a threat to public health, leading to higher morbidity and mortality rates and significant economic consequences [2]. This is because infections caused by resistant bacteria cannot be treated effectively with common antibiotics [3]. Currently, some bacteria are resistant to nearly all available antibiotics (Methicillin-resistant *Staphylococcus aureus*, carbapenemase-producing Enterobacteriaceae, extended-spectrum beta-lactamase-producing Enterobacteriaceae, extremely drug-resistant *Pseudomonas*), and as they multiply, their offspring inherit similar resistance, increasing the likelihood of untreatable infections [4]. Based on models generated from data from 204 countries, around 5 million deaths in the world were attributed to infections by multidrug-resistant bacteria in 2019 [5,6]. In Europe, these infections cause approximately 33,000 deaths annually, resulting in a cost of 1.5 billion euros, and in Spain, around 3000 deaths annually are attributed to this issue [7]. Furthermore, these figures are expected to increase dramatically over time [4,5,6].

If no action is taken, the World Health Organization (WHO) estimates that by 2050, deaths due to antimicrobial resistance could reach 10 million globally, particularly if the One Health strategy, which emphasizes the interconnectedness of human, animal, and environmental health, is not implemented [8]. Therefore, as early as 2006, the WHO set the goal for patients to receive appropriate drugs tailored to their clinical needs, at doses adjusted to their specific situations, for an appropriate duration, and at minimal cost to them and the community [9]. In pursuit of this goal, the Global Action Plan on Antimicrobial Resistance was launched in 2015, focusing on raising awareness through communication, education, and training; strengthening research; implementing infection prevention and control measures to reduce incidence; promoting rational antibiotic use in human and animal health; and managing economic resources to increase investment in drugs, diagnostics, and vaccines [10].

One of the primary drivers of rising bacterial resistance is the irrational use of antibiotics [11,12]. Addressing this issue requires enhanced training for prescribers and dispensers to ensure they are equipped to provide patients with necessary information, with community pharmacists often serving as the first point of contact for patients due to various factors [13]. Hence, understanding the preparedness of students as future healthcare professionals is crucial. The European project Student-PREPARE assessed medical students’ attitudes, knowledge, and perceptions regarding the university education they receive on antibiotics and resistance through a questionnaire [14]. There is a validated tool for future antibiotics prescribers that measures their perception of academic training on antibiotics and resistance, facilitating the identification of areas needing improvement [14]. However, pharmacy students, as future dispensers, are equally crucial healthcare professionals involved in the rational use of antibiotics [15,16]. Previous evidence shows that pharmacy students around the world exhibit deficient knowledge in the use of antibiotics and resistance [17,18]. Therefore, evaluating their knowledge, attitudes, and perceptions regarding their university education on this subject is essential. The undergraduate pharmacy curriculum in Spain spans five academic years and includes antibiotic-related training within specific courses such as microbiology, public health, microbiological and parasitological analysis, and veterinary pharmacy, which are covered from the second to the fifth years [19]. Pharmacists in the Spanish healthcare system are tasked with controlling the prescription and utilization of antibiotics at both hospital and community levels [20]. It is also known from previous studies that factors such as age or sex can influence the acquisition of knowledge and the development of certain attitudes towards health topics [21,22]. These are important factors to consider when developing training programs. Identifying and addressing gaps like this in their academic training could potentially improve future consumption indicators and antimicrobial resistance rates in Spain, which currently exceed the European average [23,24]. At present, there are no validated tools available to assess the attitudes, knowledge, and perceptions of pharmacy students regarding their education on antibiotics and resistance. Thus, our objective was to validate a questionnaire that evaluates pharmacy students’ knowledge, attitudes, and perceptions regarding their undergraduate education on antibiotics and antimicrobial resistance.

## 2. Results

### 2.1. Demographic Characteristics of Participants

In this pilot study, a total of 61 completed questionnaires were received, corresponding to 61 students (response rate to the invitation was 20.5%). All those who accepted the invitation completed and answered the questionnaire. The average age was 21.82 ± 3.81 years, with 20 males (32.8%) and 41 females (67.2%). The distribution of students’ ages by year of study can be seen in Table 1. Additionally, 4 individuals (6.6%) reported not being citizens of the country where they study pharmacy, compared to 57 (93.4%) who were. Regarding the total duration of pharmaceutical training at their home faculty, 1 person (1.6%) stated 4 years, 55 people (90.2%) stated 5 years, 1 person (1.6%) indicated 5.5 years/11 semesters, 2 people (3.3%) indicated 6 years, and 2 people (3.3%) indicated 7 years.

### 2.2. Validation and Reliability of the Questionnaire


**Step 1. Content and Face Validity of the Questionnaire**


The content validity judgment stage consisted of an appraisal by a nominal group of five experts. These five experts in questionnaire validation were professionals from the Preventive Medicine and Public Health Service of the Clinic Hospital of Santiago de Compostela, as well as professionals from the University of Santiago de Compostela with proven experience and several publications in high-impact journals on this topic. Based on the contributions from the nominal group, the following blocks of questions, which measured dimensions of the problem related to infection diagnosis and antibiotic prescription indications, were removed, as they were not applicable to university pharmacy students: (1) infection diagnosis, (2) non-prescription indications, (3) initial antibiotherapy, and (4) antibiotherapy reevaluation. Similarly, some questions were reformulated to adapt them to pharmacy students. Specifically, within the predefined dimension of “quality of care”, the question “to measure/review the use of antibiotics in the clinical setting and interpret the results of such studies” was changed to “to understand the motivation for the use of antibiotics in the clinical setting”; within the “communication skills” dimension, “to measure/review the use of antibiotics in the clinical setting and interpret the results of such studies” was changed to “to communicate to the patient the needs or limitations of antibiotic use in the clinical setting”, and “to work within the multidisciplinary team dedicated to antibiotic management in hospitals” was changed to “to communicate effectively within the multidisciplinary team dedicated to antibiotic management in hospitals”. This last question refers to the active role of the hospital pharmacist in antibiotic matters. Face validity was assessed through a focus group of pharmacy students who confirmed the questionnaire previously defined by the nominal group by consensus.


**Step 2. Reliability Analysis**



**Internal Consistency and Item Discrimination Analysis**


The initial Cronbach’s alpha value was 0.794. In this first reliability analysis, item 19: “Faculty methodology: e-learning” was removed due to a homogeneity index of 0.072. A subsequent reliability analysis excluding this item yielded a Cronbach’s alpha value of 0.80. No other item had a homogeneity index significantly <0.2, as shown in Table 2.

The following table, Table 3, shows the correlation coefficients of the item discrimination ability, all with *p* < 0.05.


**Construct Validity**


Once the reliability of the test was verified, confirmatory factor analysis was conducted with KMO = 0.656 and *p* < 0.01 for the significance of Bartlett’s test of sphericity. As shown in Table 3, the model saturated into five components that explain 75.22% of the total variance, corresponding to the five dimensions defined a priori.

In Table 4, it can be observed how each component contributes to the overall value of the scale, as well as the Cronbach’s alpha values for each component or dimension.

### 2.3. Exploratory Results of Questionnaire Responses


**Students’ Perception Regarding Quality of Care, Communication Skills, and Antibiotic Resistance**


Table 5 presents the mean and standard deviation of each item from the first three dimensions, concerning students’ perception of quality of care, communication skills, and antibiotic resistance. Additionally, student responses are categorized into disagree, neutral, and agree for clarity in interpreting the results.


**Learning Methodology and Antibiotic Training in the Faculty**


To understand pharmacy students’ perceptions regarding teaching methodology and antibiotic training at the faculty level, Table 6 displays the mean and standard deviation for each item from dimensions 4 and 5. Additionally, student responses are categorized into disagree, neutral, and agree for better comprehension of the results.

In the question “How do you think training on antibiotic treatment and prudent antibiotic use can be improved?”, collected as an open-ended response, it is noteworthy that most students suggested that training could be enhanced through more practical training, especially in clinical settings, and participatory activities such as seminars and informative talks.

Dependencies between gender and age variables were evaluated. There were no significant differences observed between the mean responses and gender regarding items assessing dimensions 1, 2, 3, and 4. However, significant differences were observed for specific items within dimension 5 (see Table 7).

Furthermore, statistically significant differences were observed between different age groups and the means obtained in the items that comprise dimension 4 of the questionnaire (Table 8). There were no statistical differences between the analyzed age groups and the other dimensions of the questionnaire.

## 3. Discussion

To our knowledge, this is the first study to validate a questionnaire measuring the knowledge, attitudes, and perceptions of pharmacy students regarding the education received during their pharmacy degree. The scale resulting from adapting the PREPARE project questionnaire for university pharmacy students demonstrated high reliability, acceptable validity indices, and good data fit. According to guidelines for developing instrumental studies, the scale meets psychometric properties, making it a reliable tool with high internal consistency and item discrimination [25]. Additionally, the questionnaire proves valid, as confirmed by the PCA.

Assessing pharmacy students’ knowledge and perceptions of their education at university is essential to identify areas for improvement and counteract antimicrobial resistance stemming from its misuse. Several scales have been developed to measure factors associated with antibiotic misuse among different healthcare professionals worldwide [26,27,28]. However, most of these scales have not been fully validated, limiting their applicability and reliability. Recent studies have developed and evaluated questionnaires aimed at measuring knowledge and attitudes towards antibiotic prescribing and bacterial resistance among physicians and the general population [29,30]. These questionnaires demonstrated content and face validity, as well as adequate reliability in terms of internal consistency and reproducibility over time. Despite this, fully validating scales designed to measure factors influencing inappropriate antibiotic use in various healthcare contexts and among different health professionals remains crucial. Those focusing on university students do not specifically address education in antibiotics and resistance, often using scales validated for the general population or other groups [31,32,33,34,35,36,37]. This lack of specificity prevents identifying areas for improvement in education.

From a public health and policy perspective, the questionnaire we developed offers several advantages over other published instruments: (i) it enables assessment, identification, and description of attitudes, perceptions, and knowledge regarding pharmacy student education on antibiotics and resistance, facilitating evaluation of teaching methodologies throughout the degree program and (ii) it can be used in different contexts, allowing assessment of knowledge and teaching methodologies across different universities. Furthermore, considering the rising rates of antimicrobial resistance and decreasing antibiotic effectiveness, the misuse of antibiotics poses a significant public health problem worldwide, threatening a return to the pre-antibiotic era [5]. Addressing deficits in pharmacy student education, as future professionals engaged in the fight against resistance, will enable the design of interventions and implementation of improvements in university education. These interventions must reflect the characteristics and barriers present in the environments where they will be implemented, making this questionnaire a suitable instrument to identify areas for improvement, a prerequisite for developing effective educational interventions and enhancing antibiotic use.

Nevertheless, we must keep in mind that the applicability of the validated questionnaire in various pharmaceutical education settings and geographic regions, particularly in low- and middle-income countries, requires careful consideration due to variations in regulations, practice standards, and the prevalence of infectious diseases. Previous studies have shown that assessment tools must be adapted to reflect specific local conditions [38,39]. In low- and middle-income countries, where health infrastructure and educational resources are often limited, the validity of the questionnaire may be compromised if it does not align with local realities. Variability in approaches to the prevention and treatment of infectious diseases, as well as differences in regulations and educational practices, suggests that the questionnaire needs to be tailored to be relevant and useful in each particular context [40,41]. For example, the topics covered and educational strategies must be reviewed and contextualized to align with current regulations and local needs. Previous research on the adaptation of questionnaires has indicated that tools must be modified to account for cultural differences and local practices to maintain their validity and usefulness [42]. Therefore, it is crucial to conduct further research to assess the reliability of the questionnaire in different contexts and to adjust assessment tools according to local characteristics and challenges. Future research should focus on implementing the questionnaire in various regions, making necessary adjustments to optimize its effectiveness in global pharmaceutical education [43].

### Questionnaire Development

Likert scales are well-established instruments for data collection across various domains, with numerous studies supporting their utility, validity, and reliability for quantifying subjective phenomena [44,45,46]. Face validity, as a form of validity referring to subjective evaluation [47], and content validity, as a measure of scale comprehensiveness and representativeness [48], were assessed and ensured by expert panels, literature review, and PCA results. High Cronbach’s alpha values obtained for 4 out of 5 components comprising the questionnaire, along with a high overall value of 0.80, indicate strong internal consistency of responses. This is corroborated by 71.7% of students disagreeing with Item 24: “Overall, do you feel you have received sufficient education at the pharmacy faculty on the use of antibiotics for your future practice”, potentially related to low scores on Dimension 4 items. Additionally, the questionnaire identifies differences in teaching methodology based on students’ gender and age variables. Previous studies show that there are differences related to sex in the acquisition of knowledge, attitudes, and practices when evaluating certain health topics, with females being more favorable towards acquiring knowledge [49,50,51]. This is also observed in our study, where women have significantly higher averages than men in various evaluated items. It has been decided to explore how age influences critical capacity concerning the methodology used in university teaching regarding antibiotics and resistance. We believe that exploring this factor with respect to age provides more information than doing so by academic year, as in the earlier years of the pharmacy degree program, much of the analyzed teaching methodology has not yet been employed. Additionally, as shown in Table 1, the older students were also in the higher academic years. Notably, older students exhibit greater use of specific methodologies like role playing and clinical case discussions or vignettes, consistent with practices in advanced pharmacy courses at the University of Santiago de Compostela. In this regard, our questionnaire is able to identify these gaps related to sex and age.

Among the study limitations, we acknowledge sample size and selection. Nonetheless, 61 participants appear reasonable for a study focusing on questionnaire development and validation as a preliminary survey or scale development [52]. Moreover, this sample size allowed confirmation of construct validity using factorial analysis like PCA. Evidence suggests that for an initial analysis of items, a sample size between 50 and 100 subjects can be sufficient. However, to achieve higher reliability, it is recommended to eventually have between 5 and 10 subjects per item, with a minimum of 300 participants [53]. To address this, we have conducted a penalized PCA using Ridge PCA, which is suitable for small sample sizes and prevents overfitting. Additionally, we applied bootstrapping techniques to ensure the robustness of our results (this analysis can be found in the Appendix A). This approach mitigates the limitation posed by our small sample size. The results of these analyses confirm our initial analysis, so we can conclude that our results are robust. Another limitation relates to the measurement scale itself. Likert-type scales often face a higher likelihood of bias [54], notably central tendency bias due to social desirability, where respondents avoid extreme scale options. We mitigated this bias by using Likert scales with more than five categories, making them akin to visual analog scales capable of discerning subtle score differences [55,56]. Another potential limitation of the study is that, as can be seen in Table 4, Factor 5 contributes a Cronbach’s alpha value of 0.30 to the overall questionnaire. Factor 5 consists of items 17, 20, and 21, which respectively have a homogeneity index of 0.284, 0.171, and 0.303 (Table 2). The only value not significantly lower than 0.2 was item 20. If this item were removed, the Cronbach’s alpha for the scale would increase from 0.80 to 0.801 (Table 2), which is a statistically insignificant change [57,58,59]. Therefore, item 20 was retained. The decision to keep an item in the questionnaire is based on the homogeneity index of the items individually, rather than on the Cronbach’s alpha of a factor in a rotated scale [57,58,60]. Thus, these items were retained. The overall Cronbach’s alpha value for the scale was 0.80, which is considered an acceptable value for a questionnaire [57,60]. Future studies should confirm these findings with larger sample sizes and assess the stability of results over time through test-retest analysis.

Strengths of our study include a systematic validation method. The questionnaire underwent review by a nominal group of experts and a focus group of six students. Items were rephrased until content clarity was ensured, securing content validity. The achieved Cronbach’s alpha ensures reliability, and correlation coefficient values indicate high internal consistency. Regarding construct validity analysis, principal component analysis confirmed and provided meaning to the predefined five dimensions, demonstrating high construct validity [57,58,59,60].

## 4. Materials and Methods

### 4.1. Study Design, Population, and Sample

The study was conducted from February 2024 to March 2024 in Galicia, a region in northwest Spain. To address the study objectives, a cross-sectional pilot study was undertaken in Santiago de Compostela.

The study universe comprised pharmacy degree students, and the study population consisted of second-year and higher students enrolled in the pharmacy faculty at the University of Santiago de Compostela. Inclusion criteria were defined as enrollment in the pharmacy degree program at the University of Santiago de Compostela, while exclusion criteria included enrollment in the first year of the pharmacy degree program at the University of Santiago de Compostela. The reason for excluding first-year students is their limited exposure to antibiotic-related education, as microbiology is part of the curriculum starting from the second year at this university [61].

### 4.2. Sample Selection 

The students were invited to participate and recruited through the dissemination of information about the study, including its objectives and significance, via the WhatsApp social network. Along with this information, a link generated on RedCap (Vanderbilt University, Nashville, TN, USA) was provided, granting access to the online questionnaire, along with a code to enter once the link was opened. The questionnaire was anonymous, and participants did not receive any form of incentive for participating [62].

### 4.3. Questionnaire Design and Validation Method

Based on the validated questionnaire used for medical students from over 20 European countries as part of the European project Student-PREPARE, promoted by ESGAP (Study Group for Antimicrobial Stewardship) of ESCMID (European Society of Clinical Microbiology and Infectious Diseases) [63], a translation into Spanish and adaptation of items for university pharmacy students was conducted. The preliminary version of the questionnaire was supplemented and completed through a literature review [31,32,33,34,35,36,37]. This version was structured into 5 dimensions with a total of 28 questions divided into 3 blocks. In Block 1, participants’ sociodemographic characteristics were collected (items 1 to 6), which were open-ended and multiple-choice. Block 2 included items measuring knowledge, attitudes, and perceptions regarding education on antibiotics and antimicrobial resistance among university pharmacy students. This Block 2 encompassed 3 different dimensions of the study. The first dimension involved students’ perception of their preparedness in quality of care (items 7 and 8); the second dimension assessed students’ perception of their preparedness in communication skills (items 9 and 10); and the third dimension focused on students’ knowledge regarding antibiotic resistance (items 11 to 14). These items were rated on a Likert scale with 9 categories. Block 3 included the fourth dimension, covering teaching methodology and students’ opinions on the methods used (items 15 to 23), rated on a Likert scale with 5 categories, and the fifth dimension, evaluating education on antibiotics within the pharmacy faculty (items 24 to 27), which were categorical response items. Similar to the original questionnaire, an additional question at the end of the questionnaire (item 28) assessed language used, also measured on a categorical scale. The questionnaire can be found as Appendix A.

### 4.4. Questionnaire Dissemination

The questionnaire was distributed to 2nd, 3rd, 4th, and 5th-year pharmacy students via the WhatsApp social network, providing a RedCap (Vanderbilt University, Nashville, TN, USA) link and access code. Three distributions were made: the first on 29 February 2024; the second on 13 March 2024; and the third on 22 March 2024.

### 4.5. Statistical Analysis

A descriptive analysis of the study sample was conducted. Normality of variables was assessed using the Kolmogorov–Smirnov test. Quantitative variables were reported as mean and standard deviation, while discrete and qualitative variables were presented as number and percentage. Variables in Block 2 concerning quality of care, communication skills, and antibiotic resistance were assessed on a Likert scale from 0 (completely disagree) to 9 (completely agree). Teaching methodology and education on antibiotics in the pharmacy faculty were transformed into a Likert scale from 0 (completely disagree) to 3 (completely agree) and subsequently categorized as “disagree”, “neutral”, and “agree” based on responses. Questionnaire validation was conducted in two steps (Figure 1) [64,65,66]:


**Step 1. Content and face validity of the questionnaire**


Content validity was evaluated through two stages: (1) development stage and judgment stage by a nominal group of experts consisting of questionnaire development specialists and university professors, which included assessment of grammar, syntax, organization, appropriateness, and logical sequence of statements [49,67]; (2) face validity was assessed via a focus group of pharmacy students to ensure consensus on item understanding in the Spanish language.


**Step 2. Reliability analysis**


Reliability analysis was conducted in two stages: (1) Evaluation of internal consistency through Cronbach’s alpha calculation, considering Cronbach’s alpha significant if >0.7 [57,60], and analysis of item discrimination capability: homogeneity of each questionnaire element was assessed to ensure alignment with the overall item set. Homogeneity index was based on Pearson correlation coefficient between item score and sum of scores from other items. Items with homogeneity < 0.2 were eliminated from the questionnaire, as they did not measure the same construct as the overall questionnaire items [57,58,59]. A significance level of *p* < 0.05 was used to reject the null hypothesis that obtained correlation was due to chance [57,60]. And (2) evaluation of the resulting model through principal component analysis (PCA) with the varimax rotation method. To select the components, we based our approach on the Kaiser Criterion, by which components with eigenvalues greater than 1 are retained. This criterion was complemented by PCA with cross-validation. Components with 2 or more items correlating >0.4 were identified as relevant [57]. Prior to factorial analysis, Kaiser–Meyer–Olkin (KMO) measure of sampling adequacy was calculated, complemented by Bartlett’s test of sphericity for verification.

### 4.6. Ethical and Legal Considerations

The study was approved by the Territorial Committee of Ethics in Research of Santiago-Lugo (registration code: 2014/386), ensuring informed consent from all participants.

## 5. Conclusions

The analysis of reliability and validity results indicates that the questionnaire discussed in this study can be applied to other populations of pharmacy students to assess their attitudes, knowledge, and perceptions regarding antibiotic use and dispensing, as well as bacterial resistance. This will facilitate the evaluation of these parameters to detect deficiencies in curricula or teaching methodologies and develop strategies and interventions to enhance the education of future antibiotic dispensers, enabling them to make rational use of antibiotics and effectively communicate information on antimicrobial resistance to patients as part of their role as healthcare professionals.

## Figures and Tables

**Figure 1 antibiotics-13-00811-f001:**
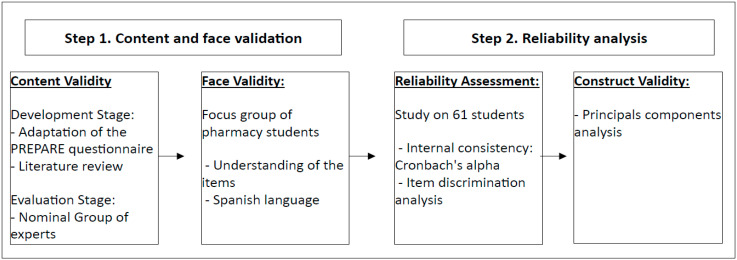
Flowchart of questionnaire development and validation.

**Table 1 antibiotics-13-00811-t001:** Distribution of students’ ages by year of study.

Year of Study	n	M	SD	Min	Max
Second year	6	18.83	0.41	18	19
Third year	9	19.56	0.53	19	20
Fourth year	16	20.88	0.5	20	22
Fifth year	30	23.6	4.76	21	46
Total	61	21.82	3.81	18	46

M: mean. SD: standard deviation.

**Table 2 antibiotics-13-00811-t002:** Scale item—total statistics.

Item (Variable)	Cronbach’s Alpha If Item Deleted	Homogeneity Index
Item 7: I feel capable of understanding the motivation for the use of antibiotics in the clinical setting	0.778	0.539
Item 8: I feel capable of working within the multidisciplinary team dedicated to antibiotic management in hospitals	0.78	0.505
Item 9: I feel capable of communicating to patients the needs or limitations of antibiotic use in the clinical setting	0.772	0.611
Item 10: I feel capable of effectively communicating within the multidisciplinary team dedicated to antibiotic management in hospitals	0.791	0.393
Item 11: I feel capable of using knowledge about common antibiotic resistance mechanisms in pathogens	0.772	0.608
Item 12: I feel capable of using knowledge about the epidemiology of bacterial resistance, including local/regional variations	0.781	0.495
Item 13: I feel capable of practicing effective infection control and hygiene (to prevent the spread of bacteria)	0.784	0.453
Item 14: I feel capable of using knowledge about the negative consequences of antibiotic use (bacterial resistance, toxic or adverse effects, cost, Clostridium difficile infections)	0.779	0.528
Item 15: Faculty methodology: lectures with fewer than 15 people	0.799	0.211
Item 16: Faculty methodology: small group teaching	0.797	0.24
Item 17: Faculty methodology: discussion of clinical cases and vignettes	0.795	0.284
Item 18: Faculty methodology: active learning tasks	0.795	0.295
Item 20: Faculty methodology: role playing	0.801	0.171
Item 21: Faculty methodology: rotation in infectious disease units	0.794	0.303
Item 22: Faculty methodology: rotation in microbiology units	0.802	0.186
Item 23: Faculty methodology: tutored or semi-tutored teaching	0.794	0.301
Item 24: In general, do you think you have received sufficient training in the pharmacy faculty on the use of antibiotics?	0.791	0.49
Item 25: Have any of the pharmacy faculty exams included questions about antibiotic treatment?	0.798	0.245

**Table 3 antibiotics-13-00811-t003:** Correlation coefficients of the scale items.

Item (Variable)	Correlation Coefficient	*p*-Value
Item 7: I feel capable of understanding the motivation for the use of antibiotics in the clinical setting	0.628	<0.01
Item 8: I feel capable of working within the multidisciplinary team dedicated to antibiotic management in hospitals	0.622	<0.01
Item 9: I feel capable of communicating to patients the needs or limitations of antibiotic use in the clinical setting	0.695	<0.01
Item 10: I feel capable of effectively communicating within the multidisciplinary team dedicated to antibiotic management in hospitals	0.525	<0.01
Item 11: I feel capable of using knowledge about common antibiotic resistance mechanisms in pathogens	0.698	<0.01
Item 12: I feel capable of using knowledge about the epidemiology of bacterial resistance, including local/regional variations	0.604	<0.01
Item 13: I feel capable of practicing effective infection control and hygiene (to prevent the spread of bacteria)	0.56	<0.01
Item 14: I feel capable of using knowledge about the negative consequences of antibiotic use (bacterial resistance, toxic or adverse effects, cost, Clostridium difficile infections)	0.617	<0.01
Item 15: Faculty methodology: lectures with fewer than 15 people	0.311	0.02
Item 16: Faculty methodology: small group teaching	0.24	<0.01
Item 17: Faculty methodology: discussion of clinical cases and vignettes	0.369	<0.01
Item 18: Faculty methodology: active learning tasks	0.356	<0.01
Item 20: Faculty methodology: role playing	0.266	0.04
Item 21: Faculty methodology: rotation in infectious disease units	0.4	<0.01
Item 22: Faculty methodology: rotation in microbiology units	0.304	0.02
Item 23: Faculty methodology: tutored or semi-tutored teaching	0.38	<0.01
Item 24: In general, do you think you have received sufficient training in the pharmacy faculty on the use of antibiotics?	0.525	<0.01
Item 25: Have any of the pharmacy faculty exams included questions about antibiotic treatment?	0.284	0.03

**Table 4 antibiotics-13-00811-t004:** Rotated component matrix and Cronbach’s alpha for the components.

Item (Variable)	Component				
	1	2	3	4	5
Item 21: Faculty methodology: rotation in infectious disease units			0.73	0.55	
Item 22: Faculty methodology: rotation in microbiology units			0.79		
Item 23: Faculty methodology: tutored or semi-tutored teaching		0.46	0.63		
Item 15: Faculty methodology: lectures with fewer than 15 people		0.84			
Item 16: Faculty methodology: small group teaching		0.82			
Item 9: I feel capable of communicating to patients the needs or limitations of antibiotic use in the clinical setting	0.87				
Item 13: I feel capable of practicing effective infection control and hygiene (to prevent the spread of bacteria)	0.83				
Item 11: I feel capable of using knowledge about common antibiotic resistance mechanisms in pathogens	0.84				
Item 7: I feel capable of understanding the motivation for the use of antibiotics in the clinical setting	0.79				
Item 14: I feel capable of using knowledge about the negative consequences of antibiotic use (bacterial resistance, toxic or adverse effects, cost, Clostridium difficile infections)	0.78				
Item 8: I feel capable of working within the multidisciplinary team dedicated to antibiotic management in hospitals	0.82				
Item 24: In general, do you think you have received sufficient training in the pharmacy faculty on the use of antibiotics?	−0.67				
Item 12: I feel capable of using knowledge about the epidemiology of bacterial resistance, including local/regional variations	0.73				
Item 25: Have any of the pharmacy faculty exams included questions about antibiotic treatment?	−0.68				
Item 18: Faculty methodology: active learning tasks					0.83
Item 10: I feel capable of effectively communicating within the multidisciplinary team dedicated to antibiotic management in hospitals	0.69				0.55
Item 17: Faculty methodology: discussion of clinical cases and vignettes				0.8	
Item 20: Faculty methodology: role playing				0.83	
**Cronbach’s Alpha**	**0.8**	**0.73**	**0.86**	**0.82**	**0.3**
**Cronbach’s Alpha of the total scale**	**0.8**				

Extraction method: PCA. Rotation method: varimax with Kaiser. Convergence was achieved in 7 iterations.

**Table 5 antibiotics-13-00811-t005:** Pharmacy students’ perceptions regarding quality of care, communication skills, and knowledge of antibiotic resistance.

Item (Variable)	M	SD	Agreement (%)	Neutral (%)	Disagreement (%)
Item 7: I feel capable of understanding the motivation for the use of antibiotics in the clinical setting	6.3	2.3	53.3	41.7	5
Item 8: I feel capable of working within the multidisciplinary team dedicated to antibiotic management in hospitals	5	2.4	27.7	55.3	17
Item 9: I feel capable of communicating to patients the needs or limitations of antibiotic use in the clinical setting	5.3	2.6	59.6	33.3	7
Item 10: I feel capable of effectively communicating within the multidisciplinary team dedicated to antibiotic management in hospitals	6.6	2.2	36	46	18
Item 11: I feel capable of using knowledge about common antibiotic resistance mechanisms in pathogens	6.1	2.4	55.2	32.8	12.1
Item 12: I feel capable of using knowledge about the epidemiology of bacterial resistance, including local/regional variations	5.2	2.4	34.5	49.1	16.4
Item 13: I feel capable of practicing effective infection control and hygiene (to prevent the spread of bacteria)	6.5	2.4	58.6	31	10.3
Item 14: I feel capable of using knowledge about the negative consequences of antibiotic use (bacterial resistance, toxic or adverse effects, cost, Clostridium difficile infections)	6.6	2.2	63.2	33.3	3.5

M: mean. SD: standard deviation.

**Table 6 antibiotics-13-00811-t006:** Pharmacy students’ perceptions regarding teaching methodology and antibiotic training at the faculty.

Item (Variable)	M	SD	Agreement (%)	Neutral (%)	Disagreement (%)
Item 15: Faculty methodology: lectures with fewer than 15 people	0.6	0.83	21.8	16.4	61.8
Item 16: Faculty methodology: small group teaching	0.96	0.68	20.4	55.1	24.5
Item 17: Faculty methodology: discussion of clinical cases and vignettes	0.91	0.83	29.6	31.5	38.9
Item 18: Faculty methodology: active learning tasks	1	0.67	22.2	55.6	22.2
Item 20: Faculty methodology: role playing	0.5	0.72	13	24.1	63
Item 21: Faculty methodology: rotation in infectious disease units	0.32	0.68	11.9	8.5	79.7
Item 22: Faculty methodology: rotation in microbiology units	0.54	0.73	14	26.3	59.6
Item 23: Faculty methodology: tutored or semi-tutored teaching	0.37	0.7	12.3	12.3	75.4
Item 24: In general, do you think you have received sufficient training in the pharmacy faculty on the use of antibiotics?	0.55	0.89	26.7	1.7	71.7
Item 25: Have any of the pharmacy faculty exams included questions about antibiotic treatment?	1.5	0.87	75	0	25

M: mean. SD: standard deviation.

**Table 7 antibiotics-13-00811-t007:** Differences between the mean responses and gender across different dimensions.

Item (Variable)	Woman (M ± SD)	Man (M ± SD)	*p*-Value
Item 7: I feel capable of understanding the motivation for the use of antibiotics in the clinical setting	6.30 ± 2.15	6.35 ± 2.68	0.94
Item 8: I feel capable of working within the multidisciplinary team dedicated to antibiotic management in hospitals	5.07 ± 2.46	4.76 ± 2.44	0.69
Item 9: I feel capable of communicating to patients the needs or limitations of antibiotic use in the clinical setting	6.57 ± 2.52	6.35 ± 2.37	0.75
Item 10: I feel capable of effectively communicating within the multidisciplinary team dedicated to antibiotic management in hospitals	5.16 ± 2.58	5.58 ± 2.59	0.58
Item 11: I feel capable of using knowledge about common antibiotic resistance mechanisms in pathogens	6.11 ± 2.27	5.95 ± 2.56	0.81
Item 12: I feel capable of using knowledge about the epidemiology of bacterial resistance, including local/regional variations	5.17 ± 2.35	5.32 ± 2.50	0.83
Item 13: I feel capable of practicing effective infection control and hygiene (to prevent the spread of bacteria)	6.53 ± 2.53	6.44 ± 2.28	0.91
Item 14: I feel capable of using knowledge about the negative consequences of antibiotic use (bacterial resistance, toxic or adverse effects, cost, Clostridium difficile infections)	6.63 ± 2.24	6.63 ± 2.27	1
Item 15: Faculty methodology: lectures with fewer than 15 people	0.42 ± 0.76	1.00 ± 0.87	0.02
Item 16: Faculty methodology: small group teaching	0.84 ± 0.72	1.18 ± 0.53	0.1
Item 17: Faculty methodology: discussion of clinical cases and vignettes	0.80 ± 0.80	0.84 ± 0.72	0.2
Item 18: Faculty methodology: active learning tasks	0.87 ± 0.62	1.29 ± 0.73	0.05
Item 20: Faculty methodology: role playing	0.38 ± 0.68	0.76 ± 0.75	0.07
Item 21: Faculty methodology: rotation in infectious disease units	0.10 ± 0.44	0.79 ± 0.86	<0.01
Item 22: Faculty methodology: rotation in microbiology units	0.38 ± 0.60	0.89 ± 0.90	0.04
Item 23: Faculty methodology: tutored or semi-tutored teaching	0.20 ± 0.52	0.76 ± 0.90	0.02
Item 24: In general, do you think you have received sufficient training in the pharmacy faculty on the use of antibiotics?	0.46 ± 0.84	0.74 ± 0.99	0.31
Item 25: Have any of the pharmacy faculty exams included questions about antibiotic treatment?	1.46 ± 0.90	1.58 ± 0.84	0.64

M: mean. SD: standard deviation.

**Table 8 antibiotics-13-00811-t008:** Differences between mean responses and age in dimension 4.

Item (Variable)	Age Group in Years		Difference of Means	Standard Error	*p*-Value
Item 15: Faculty methodology: lectures with fewer than 15 people	≤20	21–23	−0.35	0.26	0.54
		>23	−0.63	0.3	0.12
	21–23	≤20	0.35	0.26	0.54
		>23	−0.28	0.28	0.94
	>23	≤20	0.63	0.3	0.12
		21–23	0.28	0.28	0.94
Item 16: Faculty methodology: small group teaching	≤20	21–23	0.2	0.25	1
		>23	0.09	0.29	1
	21–23	≤20	−0.2	0.25	1
		>23	−0.11	0.25	1
	>23	≤20	−0.09	0.29	1
		21–23	0.11	0.25	1
Item 17: Faculty methodology: discussion of clinical cases and vignettes	≤20	21–23	−0.35	0.24	0.45
		>23	−1.08	0.28	<0.01
	21–23	≤20	0.35	0.24	0.45
		>23	−0.74	0.26	0.02
	>23	≤20	1.08	0.28	<0.01
		21–23	0.74	0.26	0.02
Item 18: Faculty methodology: active learning tasks	≤20	21–23	−0.23	0.25	1
		>23	−0.26	0.28	1
	21–23	≤20	0.23	0.25	1
		>23	−0.03	0.24	1
	>23	≤20	0.26	0.28	1
		21–23	0.03	0.24	1
Item 20: Faculty methodology: role playing	≤20	21–23	−0.14	0.21	1
		>23	−0.83	0.25	0.01
	21–23	≤20	0.14	0.21	1
		>23	−0.7	0.23	0.01
	>23	≤20	0.83	0.25	0.01
		21–23	0.7	0.23	0.01
Item 21: Faculty methodology: rotation in infectious disease units	≤20	21–23	−0.16	0.2	1
		>23	−0.58	0.24	0.06
	21–23	≤20	0.16	0.2	1
		>23	−0.42	0.22	0.19
	>23	≤20	0.58	0.24	0.06
		21–23	0.42	0.22	0.19
Item 22: Faculty methodology: rotation in microbiology units	≤20	21–23	0.13	0.23	1
		>23	−0.21	0.27	1
	21–23	≤20	−0.13	0.23	1
		>23	−0.34	0.25	0.52
	>23	≤20	0.21	0.27	1
		21–23	0.34	0.25	0.52
Item 23: Faculty methodology: tutored or semi-tutored teaching	≤20	21–23	−0.41	0.21	0.17
		>23	−0.53	0.26	0.14
	21–23	≤20	0.41	0.21	0.17
		>23	−0.12	0.23	1
	>23	≤20	0.53	0.26	0.14
		21–23	0.12	0.23	1

## Data Availability

Data availability is under petition.

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
