# Peer review of "Validity and Reliability of a Questionnaire on Attitudes, Knowledge, and Perceptions of Pharmacy Students Regarding the Training Received on Antibiotics and Antimicrobial Resistance during Their University Studies"

_antibiotics, 2024, doi:10.3390/antibiotics13090811_

Round 1

Reviewer 1 Report

Comments and Suggestions for Authors

Microbial resistance is emerging topic in public health and I believe this article will add to the body of literature in this field. The study is nicely conducted but the manuscript could be improved. Firstly, English language should be improved. Moreover, since this topic is of public health concern, literature used for manuscript could be broader.

I do believe this is interesting and valuable study, just the manuscript should be improved in order to make it more interesting to readers. 

1. What is the main question addressed by the research?

-Since microbial resistance is emerging topic in public health, all health care professionals should be knowledgeable on this topic. Since pharmacists are most accessible health care professionals, it is important their knowledge and attitudes are appropriate, and this study evaluated pharmacy students’ attitudes, knowledge and perceptions, as students will become practitioners with their formal education on this matter.  

2. What parts do you consider original or relevant for the field? What specific gap in the field does the paper address?

-I do not consider this manuscript relevant in the field. However, since this field is relevant in general, I believe this manuscript could be of interest to a broad audience.  

3. What does it add to the subject area compared with other published material?

-It added validation of questionnaire, and majority of previously published materials did not validate their surveys.  

4. What specific improvements should the authors consider regarding the methodology? What further controls should be considered?

-I do not believe any methodology improvements are needed.  

5. Please describe how the conclusions are or are not consistent with the evidence and arguments presented. Please also indicate if all main questions posed were addressed and by which specific experiments.

-I believe everything is fine on this matter.  

6. Are the references appropriate?

-Yes  

7. Please include any additional comments on the tables and figures and quality of the data.

-Figures should be improved (quality) and tables (numbers use , instead of . for decimal places).

Comments on the Quality of English Language

Needs to be proofread 

Author Response

Microbial resistance is emerging topic in public health and I believe this article will add to the body of literature in this field. The study is nicely conducted but the manuscript could be improved. Firstly, English language should be improved. Moreover, since this topic is of public health concern, literature used for manuscript could be broader.

Thank you very much for the time you spent reviewing this manuscript. And thank you very much for this positive feedback. The literature and bibliographical references have been expanded as responses to this review and by responses to the other reviewers. In the new version of the manuscript they are references 15 to 22. Marked in yellow in the version of the manuscript with Highlights.

I do believe this is interesting and valuable study, just the manuscript should be improved in order to make it more interesting to readers. 

Thank you so much. These comments encourage us to continue working on this line of research. We consider that based on the responses to the reviewers and the modifications induced by them, the work has improved in quality. Thank you so much.

What is the main question addressed by the research?-Since microbial resistance is emerging topic in public health, all health care professionals should be knowledgeable on this topic. Since pharmacists are most accessible health care professionals, it is important their knowledge and attitudes are appropriate, and this study evaluated pharmacy students’ attitudes, knowledge and perceptions, as students will become practitioners with their formal education on this matter.  

In this work, the research team has achieved answers to a field that is still little explored, which is the degree to which pharmacists can influence the improvement of indicators regarding the use of antibiotics and antimicrobial resistance. We have focused on university pharmacy students because they will be the future dispensers of antibiotics, important actors in the control of antibiotic use. Based on this premise, we wanted to know how these students perceive their academic and university training regarding this topic, with the future purpose of identifying areas for improvement in this training. And an objective way to evaluate this perception is through a questionnaire. As there was no one that would allow us to achieve this objective, we adapted an existing questionnaire (PREPARE questionnaire), and validated it for our target population.

2. What parts do you consider original or relevant for the field? What specific gap in the field does the paper address?

-I do not consider this manuscript relevant in the field. However, since this field is relevant in general, I believe this manuscript could be of interest to a broad audience.  

From our point of view, the relevance of our research for the field studied lies in the fact that we have created and validated a tool that allows us to objectively measure the perception of university pharmacy students regarding their training in the field of antibiotics and their resistances. This field constitutes one of the main public health problems currently, so early identification of deficiencies in the training of future pharmacists will contribute to better use of antibiotics in the future.

3. What does it add to the subject area compared with other published material?

-It added validation of questionnaire, and majority of previously published materials did not validate their surveys.  

Thank you very much for your observation. Indeed, and based on the bibliographic review carried out for this study, there are no validated questionnaires that allow us to measure what we intended to measure. To our knowledge, this work is the first to create and validate a tool to measure the perception of university pharmacy students about their academic training in antibiotics and resistance.

4. What specific improvements should the authors consider regarding the methodology? What further controls should be considered?

-I do not believe any methodology improvements are needed.  

Thank you very much for your feedback.

5. Please describe how the conclusions are or are not consistent with the evidence and arguments presented. Please also indicate if all main questions posed were addressed and by which specific experiments.

-I believe everything is fine on this matter.  

Thank you very much.

6. Are the references appropriate?

-Yes  

Thank you so much. References 15 to 22 have been included in the new version of the manuscript, based on the issues also raised by other reviewers. Also due to other revisions, new references 40-41 and 48-53 have been included. Thank you so much.

7. Please include any additional comments on the tables and figures and quality of the data.

-Figures should be improved (quality) and tables (numbers use , instead of . for decimal places).

Thank you so much. The figure included in the manuscript has been improved. In the tables, the comma has been replaced by the point as the decimal sign.

Reviewer 2 Report

Comments and Suggestions for Authors

Manuscript Validity and Reliability of a Questionnaire on Attitudes, Knowledge, and Perceptions of Pharmacy Students Regarding the Training Received on Antibiotics and Antimicrobial Resistance During Their University Studies by  Otero-Batán et al. deals with developing a questionnaire for pharmacy students with the purpose evaluating the education on antibiotics and AMR. In general, the paper is written nicely, and it provides a good insight on the given topic.

I have several suggestions/comments/questions regarding the manuscript.

1. The introduction is written nicely, although my comment would be that it is too general overview. If the authors decide to make some changes, I would suggest expanding the part on pharmacy student, their formal education (in Spain) on the given topic, and possibly explain the possibilities and shortcoming of pharmacy working professionals in affecting the antibiotics consumption and AMR. To the best of my knowledge (although I have an old info), Spain is one of the countries in Europe with extremely high levels of resistance. Could you make a comment how would some changes in education potentially contribute to resolving or at least reducing these leves of resistance?

2. One technical question. Your article says that the research was conducted in 2024, and the Ethical Committee gave consent in 2014. Can you please explain the 10 gap?

3. Finally, would the authors give the questionnaire as a part of supplementary file? As they have proven their survey to be valid and useful, I believe that it can further be used by other authors (for example if I decide to use it on my students).

Best of wishes in publishing your work!

Author Response

Manuscript Validity and Reliability of a Questionnaire on Attitudes, Knowledge, and Perceptions of Pharmacy Students Regarding the Training Received on Antibiotics and Antimicrobial Resistance During Their University Studies by  Otero-Batán et al. deals with developing a questionnaire for pharmacy students with the purpose evaluating the education on antibiotics and AMR. In general, the paper is written nicely, and it provides a good insight on the given topic.

Thank you very much for your kind words. This encourages us to continue working in this line of research.

I have several suggestions/comments/questions regarding the manuscript.

  1. The introduction is written nicely, although my comment would be that it is too general overview. If the authors decide to make some changes, I would suggest expanding the part on pharmacy student, their formal education (in Spain) on the given topic, and possibly explain the possibilities and shortcoming of pharmacy working professionals in affecting the antibiotics consumption and AMR. To the best of my knowledge (although I have an old info), Spain is one of the countries in Europe with extremely high levels of resistance. Could you make a comment how would some changes in education potentially contribute to resolving or at least reducing these leves of resistance?

Thank you very much. We consider this suggestion to be very timely and valuable, and it contributes to improving the scientific quality and theoretical framework of our work. Based on it, a modification has been made to the introduction. The following paragraph has been added to the introduction of the manuscript:“There is a validated tool for future antibiotic’s prescribers that measures their perception of academic training on antibiotics and resistance, facilitating the identification of areas needing improvement [14]. However, pharmacy students, as future dispensers, are equally crucial healthcare professionals involved in the rational use of antibiotics [15,16]. Previous evidence shows that pharmacy students around the world exhibit deficient knowledge in the use of antibiotics and resistance [17,18]. Therefore, evaluating their knowledge, attitudes, and perceptions regarding their university education on this subject is essential. The undergraduate pharmacy curriculum in Spain spans five academic years and includes antibiotic-related training within specific courses such as microbiology, public health, microbiological and parasitological analysis, and veterinary pharmacy, which are covered from the second to the fifth year [19]. Pharmacists in the Spanish healthcare system are tasked with controlling the prescription and utilization of antibiotics at both hospital and community levels [20]. Identifying and addressing gaps in their academic training could potentially improve future consumption indicators and antimicrobial resistance rates in Spain, which currently exceed the European average [21,22]. At present, there are no validated tools available to assess the attitudes, knowledge, and perceptions of pharmacy students regarding their education on antibiotics and resistance. Thus, our objective was to validate a questionnaire that evaluates pharmacy students' knowledge, attitudes, and perceptions regarding their undergraduate education on antibiotics and antimicrobial resistance.”

The references supporting this new paragraph are also included:

15.- De Vries E, Johnson Y, Willems B, Bedeker W, Ras T, Coetzee R, Tembo Y, Brink A. Improving primary care antimicrobial stewardship by implementing a peer audit and feedback intervention in Cape Town community healthcare centres. S Afr Med J. 2022;112(10):812-818. doi:10.7196/SAMJ.2022.v112i10.16397.

16.- Lambert M, Smit CCH, De Vos S, Benko R, Llor C, Paget WJ, Briant K, Pont L, Van Dijk L, Taxis K. A systematic literature review and meta-analysis of community pharmacist-led interventions to optimise the use of antibiotics. Br J Clin Pharmacol. 2022;88(6):2617-2641. doi:10.1111/bcp.15254.

17.- Azechi T, Sasano H, Sato K, Arakawa R, Suzuki K. Evaluation of Knowledge Regarding the Use of Antibiotics among Pharmacy Undergraduates in Japan. J Microbiol Biol Educ. 2022;23(3):e00146-22. doi:10.1128/jmbe.00146-22.

18.- Al-Taani GM, Karasneh RA, Al-Azzam S, Bin Shaman M, Jirjees F, Al-Obaidi H, Conway BR, Aldeyab MA. Knowledge, Attitude, and Behavior about Antimicrobial Use and Resistance among Medical, Nursing and Pharmacy Students in Jordan: A Cross Sectional Study. Antibiotics (Basel). 2022;11(11):1559. doi:10.3390/antibiotics11111559.

19.- Resolución de 17 de febrero de 2011, de la Universidad de Santiago de Compostela, por la que se publica el plan de estudios de Graduado en Farmacia. Boletín Oficial del Estado, núm 53, de 3 de marzo de 2011:24400-24403. Available online: https://www.boe.es/diario_boe/txt.php?id=BOE-A-2011-4076. (accessed on 18 July 2024).

20.- Sociedad Española de Farmacéuticos de Atención Primaria. Cartera de Servicios del farmacéutico de Atención Primaria. Madrid: SEFAP; 2017. 34 p. Available on line: https://www.sefap.org/wp-content/uploads/2018/01/Cartera-Servicios-FAP-Final.pdf. (accessed on 18 July 2024).

21.- European Centre for Disease Prevention and Control. Antimicrobial consumption in the EU/EEA (ESAC-Net) - Annual Epidemiological Report 2022. Stockholm: ECDC; 2023. 27p. Available on line: https://www.ecdc.europa.eu/sites/default/files/documents/AER-antimicrobial-consumption.pdf. (accessed on 18 July 2024).

22.- European Centre for Disease Prevention and Control. Antimicrobial resistance surveillance in Europe 2023 - 2021 data. Stockholm: ECDC and WHO; 2023. 186p. Available on line: https://www.ecdc.europa.eu/sites/default/files/documents/Antimicrobial%20resistance%20surveillance%20in%20Europe%202023%20-%202021%20data.pdf. (accessed on 18 July 2024).

2. One technical question. Your article says that the research was conducted in 2024, and the Ethical Committee gave consent in 2014. Can you please explain the 10 gap?

Thank you very much for your suggestion. Ethical Committee consent was granted in 2014. The Committee authorized a line of research designed to evaluate the knowledge, attitudes, and practices of key stakeholders involved in the use of antibiotics and antimicrobial resistance. From this line of research, our group has published several works in recent years. This current work is a study framed within that line of research. Therefore, the Ethical Committee's approval is still valid for this study.

3. Finally, would the authors give the questionnaire as a part of supplementary file? As they have proven their survey to be valid and useful, I believe that it can further be used by other authors (for example if I decide to use it on my students).

Thank you very much for the suggestion. Of course. The questionnaire is attached as a supplementary file. You can now view it among the supplementary documentation of our work.

Best of wishes in publishing your work!

Thank you very much! The entire research team is delighted with the contributions you have made to our work. We hope that the responses provided are to your satisfaction and help clarify your doubts. We believe that the article has improved in quality thanks to these contributions. The theoretical framework of the study is now better established. Thank you very much!

Reviewer 3 Report

Comments and Suggestions for Authors

 Thank you for the opportunity to review the manuscript titled, "  Validity and Reliability of a Questionnaire on Attitudes, Knowledge, and Perceptions of Pharmacy Students Regarding the Training Received on Antibiotics and Antimicrobial Resistance During Their University Studies. "

The problem presented by the authors of this manuscript is very important due to the global threats related to growing antibiotic resistance in populations.

The framework of this article was planned correctly.

There are some small issues that the authors should address before the manuscript can be considered for publication.

The following are my comments describing these issues.

1. Introduction

Line 46- Currently, some bacteria are resistant to nearly all available antibiotics, ….please give an example of resistant bacteria

Line 49- According to certain models, around 5 million deaths were attributed to infections by multidrug-resistant bacteria in 2019. ….please give an example of multidrug-resistant bacteria and please add where was 5 million deaths.

2. Materials and Methods

Line 84- The study was conducted from February 2024 to May 2024 in Galicia

But in Line 132-133 -  Three distributions were made: the first on February 29, 2024; 132 the second on March 13, 2024; and the third on March 22, 2024,  please explain the discrepancy in dates

Line 112- This version was structured into 5 dimensions with a total of 28 questions

In tables we have items from 7 to 25 questions and where is the rest question from 26 to 28? (demographic data of participants is from 1 to 6 items).

Line 113- In Block 1, participants' sociodemographic characteristics were collected (items 1 to 6), which were open-ended and multiple-choice…

Please show examples of questions about participant demographics with a multiple-choice answer

Line130- The questionnaire was distributed to 2nd, 3rd, 4th, and 5th-year pharmacy students, But in Line 186-  2 people (3.3%) 185 indicated 6 years, and 2 people (3.3%) indicated 7 years. please explain the discrepancy in year of study

The authors refer to the questionnaire from publications number 14, 17, but the publications do not include a questionnaire consisting of 28 items. Please explain it and which questionnaire you validated

3. Result

Line 180 - A total of 61 completed questionnaires were received…

Why is the study group so small?

Why did the authors not attempt to collect more surveys?

How many students were studying pharmacy at that time in the second, third, fourth and fifth year?

Line 183- Regarding the total duration of  pharmaceutical training at their home faculty, 1 person (1.6%) stated 4 years, 55 people (90.2%) stated 5 years, 1 person (1.6%) indicated 5.5 years/11 semesters, 2 people (3.3%) indicated 6 years, and 2 people (3.3%) indicated 7 years.

Where are the second and third year students?

I kindly ask you to explain and specify the above comments.

Author Response

Thank you for the opportunity to review the manuscript titled, "  Validity and Reliability of a Questionnaire on Attitudes, Knowledge, and Perceptions of Pharmacy Students Regarding the Training Received on Antibiotics and Antimicrobial Resistance During Their University Studies. "

 Thank you for agreeing to review it.

The problem presented by the authors of this manuscript is very important due to the global threats related to growing antibiotic resistance in populations.

Thank you very much for your observation. We are working on a line of research that aims to evaluate the factors associated with the misuse of antibiotics and antimicrobial resistance. We hope that this work can contribute to improving this current problem.

The framework of this article was planned correctly.

Thank you very much

There are some small issues that the authors should address before the manuscript can be considered for publication.

The following are my comments describing these issues.

  1. Introduction

Line 46- Currently, some bacteria are resistant to nearly all available antibiotics, ….please give an example of resistant bacteria

Thank you very much for your observation. The following information is added to the introduction:“(Methicillin-resistant Staphylococcus aureus, carbapenemase-producing Enterobacteriaceae, extended-spectrum beta-lactamase-producing Enterobacteriaceae, extremely drug-resistant Pseudomonas)”

Line 49- According to certain models, around 5 million deaths were attributed to infections by multidrug-resistant bacteria in 2019. ….please give an example of multidrug-resistant bacteria and please add where was 5 million deaths.

Thank you very much for your observation. In the answer to the previous point, examples of resistant and multiresistant bacteria are included.
Regarding the question of where those 5 million deaths occurred, it is convenient to clarify that the predictive models to which we refer in this work were generated with data from 204 countries, therefore the models refer to attributable mortality in the world to multi-resistant bacteria. Regarding this, new information is included in the introduction. Now it consists of the following:
“Based on models generated from data from 204 countries, around 5 million deaths in the world were attributed to infections by multidrug-resistant bacteria in 2019.”

  1. Materials and Methods

Line 84- The study was conducted from February 2024 to May 2024 in Galicia

But in Line 132-133 -  Three distributions were made: the first on February 29, 2024; 132 the second on March 13, 2024; and the third on March 22, 2024,  please explain the discrepancy in dates

 Thank you very much for your comment and for having detected the error in the dates. The study period was from February to March 2024. The reference to the month of May was a writing error. Thank you so much. It is corrected in the text, in the subsection of “Study Design, population and sample”. Now in both places in the text, it is stated that the study period was from February to March 2024.

Line 112- This version was structured into 5 dimensions with a total of 28 questions

In tables we have items from 7 to 25 questions and where is the rest question from 26 to 28? (demographic data of participants is from 1 to 6 items).

Indeed, the questionnaire was designed and structured in 28 questions (items). Items 1 to 6 are those that measure sociodemographic characteristics of the participants. Questions 7 to 25 are those that try to measure what we intend to measure, the knowledge, attitudes and perceptions of pharmacy students; Question 27 is an open question (How do you think training on antibiotic treatment and prudent antibiotic use can be improved?) and question 28 serves to evaluate the quality of the language used in the questionnaire. These 2 final questions were not included in the validation study, since they were not designed to measure knowledge, attitudes, and perceptions.

The questionnaire is added as supplementary material, in this way we consider that it is easier to visualize the distribution of the items.

Thank you so much.

Line 113- In Block 1, participants' sociodemographic characteristics were collected (items 1 to 6), which were open-ended and multiple-choice…

Please show examples of questions about participant demographics with a multiple-choice answer

 Thank you very much for your observation. Items 1 and 6 are open questions. By adding the questionnaire as supplementary material, it can be better visualized.

Line130- The questionnaire was distributed to 2nd, 3rd, 4th, and 5th-year pharmacy students, But in Line 186-  2 people (3.3%) 185 indicated 6 years, and 2 people (3.3%) indicated 7 years. please explain the discrepancy in year of study

 Thank you very much for your comment. Indeed, the questionnaire was distributed to students from 2nd to 5th grade. The undergraduate training in pharmacy at the University under study is 5 years long, distributed over 5 academic years. The results referred to in line 186 refer to the duration of the academic degree for each student, since there are students whose degree training lasts more than 5 years, because for any reason they must repeat subjects that they did not previously pass. That is why some students respond that its duration is 6 or 7 years.

The authors refer to the questionnaire from publications number 14, 17, but the publications do not include a questionnaire consisting of 28 items. Please explain it and which questionnaire you validated

Thank you very much for your support. The questionnaire that we validated, as explained in the methodology section, subsection “Questionnaire Design and Validation Method”, was an adaptation for pharmacy students of the questionnaire used for medical students from over 20 European countries as part of the European project Student-PREPARE, promoted by ESGAP (Study Group for Antimicrobial Stewardship) of ESCMID (European Society of Clinical Microbiology and Infectious Diseases). Starting from this questionnaire for medical students, we eliminated items that did not apply to pharmacy students and reformulated others to adapt them to university pharmacy studies. The finally validated questionnaire is the one that appears as supplementary material.

  1. Result

Line 180 - A total of 61 completed questionnaires were received… Why is the study group so small?

The study sample was 61 students because we intended to obtain preliminary results in a “pilot” sample on this questionnaire. This characteristic is included in the discussion as a possible limitation of this work.

Why did the authors not attempt to collect more surveys?

No attempts were made to collect more surveys because the invitation to participate and the questionnaire were sent 3 times to the study population, with no more responses obtained after the last sending. Therefore, we proceeded to complete our sample in order to carry out this validation pilot.

How many students were studying pharmacy at that time in the second, third, fourth and fifth year?

At the time of the study, a total of 298 students were in the courses we refer to. The response rate to the invitation was 20.5%. 100% of those who agreed to participate completed and responded to the questionnaire. This information is included in the results section.

Line 183- Regarding the total duration of  pharmaceutical training at their home faculty, 1 person (1.6%) stated 4 years, 55 people (90.2%) stated 5 years, 1 person (1.6%) indicated 5.5 years/11 semesters, 2 people (3.3%) indicated 6 years, and 2 people (3.3%) indicated 7 years. Where are the second and third year students?

 Thank you very much for your comment. Indeed, the questionnaire was distributed to students from 2nd to 5th grade. The undergraduate training in pharmacy at the University under study is 5 years long, distributed over 5 academic years. The results referred to in line 186 refer to the duration of the academic degree for each student, since there are students whose degree training lasts more than 5 years, because for any reason they must repeat subjects that they did not previously pass. That is why some students respond that its duration is 6 or 7 years.

I kindly ask you to explain and specify the above comments.

The research team hopes that the answers help clarify your doubts. Based on your critical review, we consider that the article has increased its scientific quality. Thank you very much for your review.

Reviewer 4 Report

Comments and Suggestions for Authors

I hope these insights could contribute to enhancing the manuscript

·       Lines 161-163: Please provide literature references to support this statement.

·       Table 3: The Cronbach's alpha value for Factor 5 is notably the lowest among all factors, indicating a potential need for further refinement of these items. Please discuss more on this.

·       Lines 191-192: Please elaborate on the background and expertise of the "5 experts" to demonstrate their relevance to the topic of the questionnaire.

·       Please expand the discussion the applicability of the validated questionnaire across various settings of pharmacy education and different geographic regions, particularly in low- and middle-income countries. This discussion is crucial due to the varying nature of regulations, practice standards, and the prevalence and types of infectious diseases.

Author Response

I hope these insights could contribute to enhancing the manuscript

Thank you very much. We have incorporated all the suggestions and questions into the new version of our manuscript. This has substantially improved the scientific quality of the article. Thank you again for your suggestions.

  • Lines 161-163: Please provide literature references to support this statement.

The use of the Pearson correlation coefficient to assess the homogeneity of items is a common approach in psychometrics. This method involves calculating the correlation between each item's score and the total score from all other items, which helps determine whether the item is consistent with the construct being measured by the questionnaire. Items with low correlations (often below 0.2) are typically removed because they are not sufficiently related to the overall construct and therefore do not contribute to the reliability of the scale. There is classic literature supporting this assumption, such as: Guilford JP. Psychometric Methods. 2nd ed. New York: McGraw-Hill; 1954; Nunnally JC, Bernstein IH. Psychometric Theory. 3rd ed. New York: McGraw-Hill; 1994 (ref. Number 31 in manuscript version 1.0 and ref. Number 39 in new version). More recent articles reinforce this assumption: Zijlmans EAO, Tijmstra J, van der Ark LA, Sijtsma K. Item-score reliability in empirical-data sets and its relationship with other item indices. Educ Psychol Meas. 2018;78(6):998–1020. doi:10.1177/0013164417728358.

New references supporting this statement are included in the manuscript. Thank you very much.

  • Table 3: The Cronbach's alpha value for Factor 5 is notably the lowest among all factors, indicating a potential need for further refinement of these items. Please discuss more on this.

Thank you very much for your appreciation. Table 3 shows the Cronbach's alpha value that each factor contributes to the overall questionnaire. Factor 5 contributes a Cronbach's alpha of 0.3 to the overall questionnaire. This factor 5 consists of items 17, 20, and 21, which have homogeneity indices of 0.284, 0.171, and 0.303, respectively. The only value below 0.2 is item 20. If this item were removed, the Cronbach's alpha for the scale would increase from 0.80 to 0.801, which is a statistically insignificant change. Therefore, item 20 was retained. The decision to keep an item in the questionnaire is based on the homogeneity index of the items individually, rather than on the Cronbach's alpha of a factor in a rotated scale. Thus, these items were retained. The overall Cronbach's alpha value for the scale is 0.80, which is considered an acceptable value for a questionnaire. [Nunnally JC, Bernstein IH. Psychometric Theory. 3rd ed. New York: McGraw-Hill; 1994].

The following information is included in the discussion section: “Another potential limitation of the study is that, as can be seen in Table 3, Factor 5 contributes a Cronbach's alpha value of 0.30 to the overall questionnaire. Factor 5 consists of items 17, 20, and 21, which respectively have homogeneity index of 0.284, 0.171, and 0.303 (Table 1). The only value not significantly lower than 0.2 was item 20. If this item were removed, the Cronbach's alpha for the scale would increase from 0.80 to 0.801 (Table 1), which is a statistically insignificant change. Therefore, item 20 was retained. The decision to keep an item in the questionnaire is based on the homogeneity index of the items individually, rather than on the Cronbach's alpha of a factor in a rotated scale. Thus, these items were retained. The overall Cronbach's alpha value for the scale was 0.80, which is considered an acceptable value for a questionnaire [38,39].

  • Lines 191-192: Please elaborate on the background and expertise of the "5 experts" to demonstrate their relevance to the topic of the questionnaire.

Thank you very much for your appreciation. The nominal group consisting of 5 experts in questionnaire validation were professionals from the Preventive Medicine and Public Health Service of the Hospital Clínico de Santiago de Compostela, as well as professionals from the University of Santiago de Compostela with proven experience and several publications in high-impact journals on this topic, and who are part of the GREPHEPI Group.

In this link to PubMed, you can find the articles published by this group, of which I am a member, related to questionnaire validation and the assessment of knowledge, attitudes, and practices in specific populations involved in antibiotic use and its future improvement:
https://pubmed.ncbi.nlm.nih.gov/?term=GREPHEPI+Group&sort=pubdate

Information about the nominal group's experience in this field is included in the "Validation and Reliability of the Questionnaire" subsection of the results section. “This 5 experts in questionnaire validation were professionals from the Preventive Medicine and Public Health Service of the Hospital Clínico de Santiago de Compostela, as well as professionals from the University of Santiago de Compostela with proven experience and several publications in high-impact journals on this topic.”

  • Please expand the discussion the applicability of the validated questionnaire across various settings of pharmacy education and different geographic regions, particularly in low- and middle-income countries. This discussion is crucial due to the varying nature of regulations, practice standards, and the prevalence and types of infectious diseases.

Thank you very much for the suggestion. Based on the reviewer's input, a new paragraph has been included in the discussion section:“Nevertheless, we must keep in mind that the applicability of the validated questionnaire in various pharmaceutical education settings and geographic regions, particularly in low- and middle-income countries, requires careful consideration due to variations in regulations, practice standards, and the prevalence of infectious diseases. Previous studies have shown that assessment tools must be adapted to reflect specific local conditions [48,49]. In low- and middle-income countries, where health infrastructure and educational resources are often limited, the validity of the questionnaire may be compromised if it does not align with local realities. Variability in approaches to the prevention and treatment of infectious diseases, as well as differences in regulations and educational practices, suggests that the questionnaire needs to be tailored to be relevant and useful in each particular context [50,51]. In example, the topics covered and educational strategies must be reviewed and contextualized to align with current regulations and local needs. Previous research on the adaptation of questionnaires has indicated that tools must be modified to account for cultural differences and local practices to maintain their validity and usefulness [52]. Therefore, it is crucial to conduct further research to assess the reliability of the questionnaire in different contexts and to adjust assessment tools according to local characteristics and challenges. Future research should focus on implementing the questionnaire in various regions, making necessary adjustments to optimize its effectiveness in global pharmaceutical education [53].”

Authors believe that, based on this input and the others, the work has gained in scientific quality and external validity. Thank you very much.

Round 2

Reviewer 2 Report

Comments and Suggestions for Authors

Authors have made the suggested changes and answered my question, therefor, I believe that paper is now suitable for publishing.

Author Response

Authors have made the suggested changes and answered my question, therefor, I believe that paper is now suitable for publishing.

Thank you. The article has truly improved in scientific quality due to the reviewers' contributions. Thank you very much.

Reviewer 3 Report

Comments and Suggestions for Authors

 The authors precisely responded to the reviewer's comments.

The manuscript was corrected.

Due to the small sample size, it should add one or two sentences emphasizing that this is a preliminary study in the session: materials/ methods or results.

The authors explained everything at the end of the article, and it is correct, but           it should have included this note even earlier.

Author Response

The authors precisely responded to the reviewer's comments. The manuscript was corrected.

Thank you very much. The authors are aware that the reviewers' contributions have greatly improved the scientific quality of our work.

Due to the small sample size, it should add one or two sentences emphasizing that this is a preliminary study in the session: materials/ methods or results. The authors explained everything at the end of the article, and it is correct, but it should have included this note even earlier.

Thank you very much for the suggestion. The following sentences have been added:

1.- In the "Study Design" subsection of the "Materials and Methods": “To address the study objectives, a cross-sectional pilot study was undertaken in Santiago de Compostela.”

And in the "Sociodemographic Characteristics of Participants" subsection of the Results: In this pilot study, a total of 61 completed questionnaires were received, corresponding to 61 students […].”

This characteristic of the pilot study, as you mentioned, has been included as a limitation of the study. We are glad you think it is well done.

Thank you very much.